# Relationship between Sensory Processing Skills and Feeding Behaviors in Children Aged 3–6 Years with Cerebral Palsy with Cerebral Visual Impairment

**DOI:** 10.3390/children10071188

**Published:** 2023-07-09

**Authors:** Mustafa Cemali, Özge Cemali, Ayla Günal, Serkan Pekçetin

**Affiliations:** 1Department of Occupational Therapy, Faculty of Health Sciences, Lokman Hekim University, 06510 Ankara, Turkey; 2Department of Nutrition and Dietetics, Faculty of Health Sciences, Gazi University, 06490 Ankara, Turkey; ozgecemali@gazi.edu.tr; 3Department of Physiotherapy and Rehabilitation, Faculty of Health Sciences, Tokat Gaziosmanpaşa University, 60250 Tokat, Turkey; ayla.gunal@gop.edu.tr; 4Department of Occupational Therapy, Faculty of Gülhane Health Sciences, University of Health Sciences, 06010 Ankara, Turkey; serkan.pekcetin@sbu.edu.tr

**Keywords:** cerebral visual impairment, cerebral palsy, sensory integration, feeding

## Abstract

The current study aimed to examine the relationship between sensory processing skills and feeding behavior in cerebral palsy (CP) children aged 3–6 years with cerebral visual impairment (CVI). A total of ninety mothers participated in the study in three groups: thirty mothers of children with CP with CVI, thirty mothers of children with CP without CVI, and thirty mothers of children with typical development (TD). The sensory processing skill of the children was evaluated with the Sensory Profile (SP), and feeding behavior was evaluated with the the Behavioral Pediatric Feeding Assessment Scale (BPFAS). In the triple comparison, a significant difference was found between the groups in all SP parameter and BPFAS scores (*p* < 0.001). Post hoc analysis revealed statistically significant differences between the groups in all parameters (*p* < 0.001). Feeding problems were detected in 65% of all groups. In the correlation analysis, a significant relationship was found between all parameters of the SP and the BPFAS (*p* < 0.05). In terms of sensory processing skills and feeding status, it was determined that children with CVI with CP had more problems than children with CP without CVI, and children with CP without CVI had more problems than children with TD. With these results, it was concluded that sensory processing problems affect feeding status, and visual impairment causes both sensory problems and feeding problems.

## 1. Introduction

Cerebral palsy (CP) is a non-progressive motor function, posture, and movement disability that occurs as a result of prenatal, natal, and postnatal harm to the developing brain [1]. The main expression of the disease is motor dysfunction, but since the existing pathology affects other parts of the brain, motor disorder is often accompanied by epilepsy, sensory processing problems, visual and hearing disorders, oral–motor insufficiency and related feeding problems, orthopedic disorders, mental retardation, behavioral disorders, language–speech disorders, chronic lung problems, and sleep problems [2,3].

It is known that problems in children with CP might differ depending on the location of brain damage. Feeding problems are said to be more important than other problems since they are directly tied to growth and development [4,5]. Feeding problems in children with cerebral palsy vary depending on impaired motor coordination, structural anatomical pathologies, spasticity or hypotonic posture, sensory processing problems, motor involvement severity, whether the swallowing reflex is developed to a normal level or not, trunk control deficiencies, and deficiencies in tongue, lip, jaw, and mouth functions [6].

Feeding problems affect 60–90% of children with CP and can begin in infancy and last throughout life [7]. Feeding problems that affect physical growth can be seen in children with CP due to reasons such as difficulty in sucking, chewing, and swallowing, loss of appetite, rejection of new foods, taking a long time to eat. If these feeding problems are not treated, they can cause growth and developmental retardation, which negatively affect morbidity and mortality [8,9].

There have been many studies in the literature on the factors that cause feeding problems in children with CP [10,11,12]. Taş revealed that feeding behavior problems increased with worsening gross motor skills in children with CP aged 2–15 years [10]. Kürklü, in his study on children with CP aged 3–18, concluded that swallowing and chewing difficulties are associated with oral–motor insufficiency and that this is the most important cause of feeding problems [11]. Kangalgil, on the other hand, stated that behaviors such as keeping food in the mouth, choking while feeding, and food refusal are factors that cause feeding problems in children with CP aged 7–17 [12]. Although studies have shown that motor, swallowing, and chewing issues are the most common causes of feeding problems [4,13,14], research has also shown that sensory processing impairments can induce feeding problems related to the manipulation of solid foods in the mouth [15]. It has been reported that eating problems due to sensory reasons may result in oral intolerance, primarily due to the flavor, texture, and hardness of the meal, and that this circumstance also prolongs feeding times [16].

It is known that loss of function may occur as a result of brain damage due to CP due to motor, cognitive, behavioral, speech, auditory, and visual areas being affected. Although motor involvement is prominent in the loss of function in children with CP, cerebral visual impairment (CVI), damage to the part of the cerebral cortex that governs vision, is also indicated to cause serious problems for the functional independence of children [17]. CVI is defined as a functional visual impairment that occurs due to damage to the visual center. CVI occurs in 40 to 48% of children with CP and occurs in addition to CP symptoms [18]. Around 70–80% of the receptors in our body are in our eyes, and the sense of sight is of great importance to the integration of visual stimuli and environmental orientation [19]. Sensory processing, which is the reception of stimuli from the environment, their interpretation in the cerebral cortex, and the formation of an appropriate response, is impaired in these children with CVI and CP due to the involvement of sensory areas [20].

In studies examining the parameters associated with sensory processing skills in children with CP with and without CVI, it is seen that mostly motor skills are evaluated [19,20,21,22]. Cemali found that the vestibular, tactile, and proprioception senses of children with CVI and CP were more problematic than children with CP without CVI in his study of 10–18-month-old children with and without CVI, stating that vision loss may prevent children from perceiving sensory stimuli and that children may perceive stimuli to be dangerous [19]. Pavao evaluated the sensory parameters of children with CP with the Dunn Sensory Profile questionnaire, which included oral assessment of sub-parameters, and partially examined feeding status through feeding questions in the oral assessment and revealed that the children had feeding problems [23]. Apart from these studies, studies with children with CP are mostly related to the relationship between feeding and motor skills, swallowing, and chewing. [24,25]. It is known that the characteristics of foods, such as texture, taste, smell, and appearance, constitute the sensory parameters of feeding [26]. We could not identify a study that looked at the association between sensory processing skills and feeding status in children with CP, either with or without CVI, in our examination of the literature. From this point of view, we aimed to compare the sensory processing skills and feeding status of children with CVI with CP, children without CVI with CP, and children with TD to reveal how sensory processing skills and feeding status are related in each group.

## 2. Materials and Methods

### 2.1. Study Design

The study was approved by the Tokat Gaziosmanpasa University Clinical Research Ethics Committee (Registration Number: 01-08/Approval Date: 21 March 2023) and was carried out in accordance with the ethical rules established according to the Declaration of Helsinki. Research was carried out between March 2023 and May 2023 in a special education and rehabilitation center in Ankara where children with visual, physical, and mental disabilities receive special education and rehabilitation services.

### 2.2. Procedure

Three independent groups consisting of mothers of children aged 3–6 years with CVI with CP, mothers of children without CVI with CP, and mothers of children with TD were formed for the study. After the purpose of the study was explained to the mothers who voluntarily agreed to participate in the study, they were asked to sign a consent form. The interviews were made face-to-face and evaluation forms were filled using paper and pencils. The demographic information form, the Sensory Profile (SP), and the Behavioral Pediatric Evaluation Scale (BPFAS) were used in the evaluations made in the presence of a physiotherapist and feeding and dietetics specialist with 10 years of experience. Each participant’s evaluation session lasted an average of 30 min. After calculating the evaluation scores for each group, the SP and BPFAS scores of the groups were compared. Correlation analysis was used separately for each group to examine the relationship between the SP and the BPFAS.

### 2.3. Participant

The sample size of the study was calculated with an a priori type of power analysis by selecting the ANOVA fixed effect, omnibus, and one-way test parameter under the F test family using the Gpower 3.1.9.7 program. In order to determine the difference between the 3 different groups, the Behavioral Pediatric Feeding Assessment Scale (BPFAS) was determined for 30 individuals in each group with a medium effect size, a 95% confidence interval, and 80% power, and the required number for the total sample size was 90.

Children with CP with and without visual impairment who participated in the study were selected according to the Special Needs Report for Children Report (SNRCR). This report is a report that states the diagnosis and training needs of children, which is required to receive special education and rehabilitation services in Turkey and given by a multidisciplinary team.

The inclusion criteria for the children with CP whose mothers were included were as follows: (1) age from 36 to 72 months and (2) diagnosed with CP according to the SNRCR (for the group of children with CP with CVI, a diagnosis of CVI according to the SNRCR and no ocular problems was also required). The inclusion criteria for children with TD were as follows: (1) age of 36 to 72 months and (2) the absence of any ocular or neurological disorders. Exclusion criteria for the study were as follows: (1) the mother’s refusal to participate in the study and (2) the child being affected by deafness, blindness, or deaf-blindness. Inclusion criteria for mothers were the ability to read and write, being at a cognitive level capable of perceiving and answering the survey questions, and agreeing to answer the survey questions completely.

### 2.4. Measure

#### 2.4.1. Demographic Information Form

This is an information form that contains information such as the age and gender of the child.

#### 2.4.2. Sensory Profile (SP)

The Sensory Profile was developed by Dunn et al. in 1999 [27].to evaluate responses to possible sensory experiences in daily life in children with different diagnoses. This profile allows for evaluation of the effect of the sensory processes of individuals on functional performance in their daily lives. The profile reveals the sensory appearance of the individual in terms of sensory seeking, emotional response, low endurance, oral sensitivity, distraction, poor perception, sensory sensitivity, activity level, and fine motor/perceptual differences. The profile is filled by caregivers. A five-point Likert scale is used to evaluate the profile consisting of 125 items, with “Always” as 1; “Often” as 2; “Sometimes” as 3; “Rarely” scored as 4; and “Never” scored as 5. After the scores are calculated, a result is obtained about the child’s sensory system processing problem, how they interpret the sense, how they adapt to the surrounding sensory stimuli, how they participate in the activities of daily living, or how they react to them according to the reference score table. In addition, it evaluates the child’s ability to receive the senses (sensory processing), adjust (modulation), and create behavioral and emotional responses according to the reference scoretable. Typical performance, potential difference, and precise difference score ranges were determined for each parameter. The overall score for each parameter improves from the exact difference score range to the typical performance score range. The Sensory Profile is a frequently used questionnaire to evaluate sensory processing skills in children with CP [23]. The Turkish version of this study was conducted by Kayihan et al. in 2015, and the total cronbach α value was 0.99 [27,28].

#### 2.4.3. The Behavioral Pediatric Feeding Assessment Scale (BPFAS)

The scale, which was developed by Crist et al. [29] and adapted into Turkish by Önal et al. [30], consists of 35 items. In total, “25 of the expressions in the 5-point Likert-type scale are related to the feeding status of the child and 10 of them are related to the person responsible for the feeding of the child”. Of the 25 items of the BPFAS related to children, 6 of them (items 1, 3, 5, 6, 9, and 16) have a negative meaning. Items that are expressed positively are scored in the opposite way. The lowest score that can be obtained from the BPFAS is 35, and the highest score is 175. A total score of more than 84 indicates that there is an eating problem. There are four subheadings in the scale, namely food selectivity, early food rejection, early grainy food rejection, and late food rejection. An increase in the score obtained from the scale indicates a high level of problematic eating behavior and eating habits. The scale is a tool used to evaluate eating behavior in children with CP [10]. The total Cronbach α value of the scale is 0.87 [31].

### 2.5. Statistical Analysis

Data were analyzed using the SPSS 25.0 program (SPSS Inc. Chicago, IL, USA). Descriptive statistics of data were given as number and percentage for categorical variables and mean and standard deviation for numerical variables. The conformity of continuous variables to normal distribution was tested using the Shapiro–Wilk test. It was determined that SP and BPFAS scores did not show normal distribution; therefore, non-parametric tests were used. The differences between the outcome measures of the groups were analyzed with the chi-square test for categorical variables and the Kruskal–Wallis test for numerical variables. The Kruskal–Wallis test was used to compare the SP and BPFAS scores between the groups. Multiple comparisons were then made with the Mann–Whitney U test with Bonferroni correction [32]. The relationship between SP and BPFAS scores was also analyzed with the Spearman correlation test. A significance level of 0.05 was accepted for the *p*-value.

## 3. Result

The groups were comparable in term of gender and age (*p* > 0.05) (Table 1).

It was observed that there was a statistically significant difference between the BPFAS scores between the groups (*p* < 0.001). It was concluded that the feeding status of children with CVI with CP was more problematic than those with CP without CVI, and the feeding status of children with CP without CVI was more problematic than those with TD (Table 2).

In the assessment of sensory processing skills, a significant difference was found between the groups in all areas of the SP (*p* < 0.001). It was concluded that in all sub-areas of the SP, the sensory processing skills of children with CP with CVI were lower than those of children with CP without CVI, and the sensory processing skills of children with CP without CVI were lower than those of children with TD (Table 2).

As a result of post hoc analysis, there was a significant difference in all pairwise comparisons of the three groups in all parameters. Children with CP with CVI and CP without CVI were found to exhibit differences in all sub-areas of the BPFAS and SP (Table 3).

According to BPFAS, feeding behavior problems were detected in 10 children with TD, 22 children with CP without CVI, and 27 children with CP with CVI. There was a significant difference between the groups (*p* < 0.001) (Table 4).

There was a statistically negative correlation between all sub-parameters of the SP and BPFAS for each group (*p* < 0.05). It was concluded that with the increase in sensory processing skill problems, feeding behavior problems increased (Table 5).

## 4. Discussion

The current study studied the association between sensory integration and feeding behavior in children with CVI with CP, children without CVI with CP, and children with TD and compared these parameters between groups. In conclusion, it has been observed that children with CP with CVI have significant sensory and feeding problems compared to both children with CP without CVI and children with TD. It has been found that feeding problems increase with an increased level of sensory processing disorder.

In a study evaluating sensory processing in children with visual impairment, it was found that children have low sensory thresholds and exhibit sensitivity avoidance behavior to stimuli [33]. Although our results were similar, we found that in addition to low sensory threshold, avoidance, and sensitivity behavior, sensory seeking response also improved in children with CP with CVI. It is known that children with CP with CVI and children with CP without CVI due to neural involvement have lower sensory, motor, and cognitive skills than children with TD, and their functional independence levels are lower accordingly [34,35]. Children with functional disabilities may exhibit sensory seeking because they do not have adequate access to sensory stimuli. On the contrary, those with CP who cannot adequately access sensory stimuli may exhibit behaviors such as sensitivity and avoidance due to insufficient sensory experience [36]. It is known that vision is an important sense for environmental orientation, and it is predicted that CP symptoms may be more severe in those with vision deficiencies [17,37,38]. Therefore, it can be inferred that the response of children with CP and CVI to sensory stimuli may be variable and may produce different types of responses depending on the sensory threshold [19].

It is recognized that motor, cognitive, visual, speech, and hearing parts of the cerebral cortex, as well as the somatosensory area and sensory pathways, may be damaged in CP. In CP, the injured central nervous system not only causes aberrant muscle tone, but may also lead to sensory processing disorders arising in addition to these symptoms [39]. In a study conducted in children with CP, it was stated that 90% of the children had vestibular and proprioceptive sensory dysfunctions, and vestibular sense and motor function development were correlated bidirectionally [40]. In the current study, the vestibular sensory evaluation results of children with CP without CVI were similar to those in the literature, but this loss was much higher in children with CP and CVI. Cemali concluded that children with CP with CVI had lower motor skills and vestibular sensory development than children with CP without CVI, and revealed that this situation resulted from limitations on independent movement and environmental orientation as a consequence of visual impairment [20]. It can be said that visual and vestibular senses are related to each other and vision is an important parameter for movement [41]. Recognition of objects occurs with the cooperation of vision and tactile senses [42]. In a study conducted with children with CP, it was stated that children have sensory modulation disorders due to neural activity, and it was concluded that this disorder causes children to form a defensive response to tactile stimuli [43]. When tactile stimuli do not integrate with vision in children with visual impairment, stimuli from the environment may be regarded as dangerous, and children may avoid tactile stimuli and objects as a result. The same is true for auditory stimuli. In cases where the visual sense is insufficient, sounds of different types and intensities from the environment can be defined as a danger for children [19]. Similar to the literature, we concluded that children with CP with CVI have problems in tactile and auditory sensory processing. These results clearly show that sensory problems can be experienced in the absence of the visual sense required to perceive the environment and objects [44].

Studies have shown that the sensory integration system is linked to the limbic system, where the emotional state is controlled [45]. For this reason, after the age of three, the sensory need is more pronounced and this deficiency may cause emotional, behavioral, and social problems in a disease group such as those with CP who have a low level of independence [23]. In a study conducted with children with CP, it was concluded that children who do not experience enough sensory satisfaction are more aggressive, experience emotional problems, and engage in self-harming behaviors [36]. Another study found that sensory processing impairment limits children’s activity and participation [46]. In current research, we found that children have problems in behavioral, sensory, and social participation areas related to sensory integration. It can be said that children with neurodevelopmental disorders who cannot experience sensory stimuli adequately and experience sensory satisfaction may exhibit different behaviors, be aggressive, and experience limitations in their level of activity [21,23]. 

Behavioral, emotional, and social areas of sensory processing are as important as the vision, hearing, touch, and vestibular sensory areas, and these areas interact with each other [47]. Feeding behavior is another significant indicator that demonstrates the relationship between feeding’s tactile and behavioral characteristics. Although tactile stimuli function in the mouth as they do throughout the body, they are more sensitive in the mouth [16]. It has been stated that children with CP may create a disorganized response to tactile stimuli, with tactile stimuli in the oral region thus possibly being affected by this situation, which may affect food intake for foods with different textures and flavors [48]. It has been reported that 25–45% of infants and children with TD and 80% of children with neurodevelopmental disorders have feeding problems such as eating less, refusal to eat, choosing food, avoiding food, and prolonged eating times [49]. Yilmaz stated in his study that children with TD had moderate eating disorders, and Aydin stated that infants with TD had a low level of eating problems. İnan reported that 39.5% of children with learning difficulties had feeding problems, with Crasta finding that 61% of children with autism had feeding problems. Studies have emphasized that the most important cause of feeding disorders in children with TD is being picky with food due to the taste, texture, and smell of food, while children with neurodevelopmental disorders also have a lack of sensory tolerance [50,51,52,53]. In the current study, we found that one-third of the children with TD had eating disorders, and this was much higher in children with CP with and without CVI; with the increase in sensory problems in the groups, the eating problems increased. In a study conducted with children with CP, it was revealed that children have problems in chewing and tolerating solid foods [48]. In another study investigating feeding problems in children with CP, it was stated that oral sensory problems in addition to oromotor and neurological swallowing and chewing problems affect children’s feeding [4]. Similar to prior studies, it was established in a study of infants with a history of premature birth that the feeding status of children with sensory integration disorder was worse [54]. In the current study, we concluded that the feeding attitudes of children with CVI and CP were worse and that this was related to sensory processing skills. We think that this situation is related to the inability to tolerate the type, taste, and texture of foods, together with sensory reasons [55]. Although we focused on the relationship of feeding with sensory processing skills and oral and tactile stimuli in our study [56,57], it should be considered that motor, oral motor, swallowing, and chewing disorders also affect feeding in children with CP with and without CVI. With this perspective, it is thought that the multidimensional evaluation of feeding will be an important step in determining the feeding problem and that our findings will guide further studies to be conducted more comprehensively. 

This study was limited in that it could not directly analyze sensory status or feeding behavior, as assessments were made using standard caregiver-filled questionnaires. In addition, other limitations of the study are that other factors that may affect feeding behavior were not evaluated, such as the motor, swallowing, and cognitive states of the children and the ability of the parents to feed the children.

## 5. Conclusions

It was observed that children with CP with CVI had more problematic sensory processing skills and feeding behavior attitudes than children with CP without CVI, and children with CP without CVI had more problematic experiences than children with TD. In addition, it was concluded that as the problems in sensory processing skills increased in each group, the feeding behavior problems also increased. As a result, it was observed that the sensory processing skills and feeding behaviors of children with CP were affected, and with the addition of a CVI diagnosis to the diagnosis of CP, the sensory and feeding status of children became worse.

## Figures and Tables

**Table 1 children-10-01188-t001:** Descriptive characteristics of the groups.

	TD	CP without CVI	CP with CVI	
	(*n* = 30)	(*n* = 30)	(*n* = 30)	
Gender	*n (%)*	*n (%)*	*n (%)*	*p*
Boy	16 (53.3)	16 (53.3)	18 (60)	0.0837 ^a^
Girl	14 (46.7)	14 (46.7)	12 (40)	
	**M (SD)**	**M (SD)**	**M (SD)**	** *p* **
**Age (month)**	44.97 (7.68)	44.80 (7.51)	47.62 (9.57)	0.378 ^b^

CVI: cerebral visual impairment; CP: cerebral palsy; TD: yypical development. ^a^ The *p*-value of the chi-qquare test. ^b^ The *p*-value of the Kruskal–Wallis test.

**Table 2 children-10-01188-t002:** Comparison of groups in terms of BPFAS and SP scores.

	TD	CP without CVI	CP with CVI	
	(*n* = 30)	(*n* = 30)	(*n* = 30)	
	M (SD)	M (SD)	M (SD)	*p*
**BPFAS**	74.90 (24.4)	92.56 (19.01)	109.96 (22.26)	<0.001
**SP**				
Sensation seeking	97.50 (21.43)	81.63 (18.44)	66.70 (10.45)	<0.001
Sensation avoidance	112.56 (24.83)	91.36 (17.77)	78.26 (14.57)	<0.001
Sensory sensitivity	74 (17.78)	61.13 (12.11)	49.63 (8.18)	<0.001
Low registration	57.16 (11.98)	46.53 (10.22)	36.83 (7.24)	<0.001
Auditory processing	28.46 (7.68)	24.40 (6.40)	18.76 (4.56)	<0.001
Visual processing	32.36 (9.18)	27.63 (7.42)	23.80 (5.28)	<0.001
Vestibular processing	39.66 (9.78)	34.70 (7.43)	28.96 (4.74)	<0.001
Touch processing	65.30 (16.62)	56.3 (11.53)	48.93 (8.07)	<0.001
Oral sensory	44.53 (11.33)	35.9 (8.15)	30.4 (6.37)	<0.001
Sensory inputs that influence emotional responses	25.50 (3.88)	15.53 (3.54)	13.53 (2.93)	<0.001
Regulation of visual stimuli affecting emotional responses and activity level	15.16 (3.39)	18.60 (4.27)	10.86 (2.33)	<0.001
Emotional–social responses	56.13 (12.11)	43.9 (9.52)	36.26 (8.32)	<0.001
Behavioral consequences of sensory processing	23.16 (5.27)	18.7 (4.16)	15.8 (3.92)	<0.001
Response threshold results	11.63 (2.74)	9.46 (2.11)	7.63 (2.10)	<0.001

CVI: cerebral visual impairment; CP: cerebral palsy; TD: typical development. The *p*-values are taken from the Kruskal–Wallis test.

**Table 3 children-10-01188-t003:** Pairwise comparisons of outcome measures between groups.

	TD CP without CVI	TD CP with CVI	CP without CVI CP with CVI
	*n* = 30	*n* = 30	*n* = 30
	*p*	*p*	*p*
**BPFAS**	0.006 *	<0.001	<0.001
**SP**			
Sensation seeking	<0.001	<0.001	<0.001
Sensation avoidance	<0.001	<0.001	<0.001
Sensory sensitivity	0.001 *	<0.001	<0.001
Low registration	<0.001	<0.001	<0.001
Auditory processing	0.045 *	<0.001	<0.001
Visual processing	0.024 *	<0.001	<0.001
Vestibular processing	0.034 *	<0.001	<0.001
Touch processing	0.019 *	<0.001	<0.001
Oral sensory	0.001 *	<0.001	<0.001
Sensory inputs that influence emotional responses	0.002 *	<0.001	<0.001
Regulation of visual stimuli affecting emotional responses and activity level	0.001 *	<0.001	<0.001
Emotional–social responses	<0.001	<0.001	<0.001
Behavioral consequences of sensory processing	<0.001	<0.001	<0.001
Response threshold results	<0.001	<0.001	<0.001

CVI: cerebral visual impairment; CP: cerebral palsy; TD: typical development. All significance values have been adjusted by the Bonferroni correction. * *p* < 0.05.

**Table 4 children-10-01188-t004:** Number of children with behavioral feeding problems according to BPFAS score.

	TD		CP without CVI		CP with CVI		Total		*p*
	*n*	%	*n*	%	*n*	%	*n*	%	
Has a feeding behavior problem	10	33.3	22	73	27	90	59	65	*p* < 0.001
No feeding behavior problems	20	66.7	8	27	3	10	31	35	

CVI: cerebral visual impairment; CP: cerebral palsy; TD: typical development. BPFAS > 84: has a feeding behavior problem.

**Table 5 children-10-01188-t005:** Examining the relationship between the SP and BPFAS.

BPFAS								
	TD		CP without CVI		CP with CVI		Total	
	r	*p*	r	*p*	r	*p*	r	*p*
SP								
Sensation seeking	−881	*p* < 0.001	−735	*p* < 0.001	−402	0.020 *	–0.813	*p* < 0.001
Sensation avoidance	−882	*p* < 0.001	−761	*p* < 0.001	−475	0.008 *	–0.723	*p* < 0.001
Sensory sensitivity	−915	*p* < 0.001	−811	*p* < 0.001	−561	0.001 *	–0.717	*p* < 0.001
Low registration	−864	*p* < 0.001	−756	*p* < 0.001	−440	0.015 *	–0.698	*p* < 0.001
Auditory processing	−932	*p* < 0.001	−879	*p* < 0.001	−593	0.001 *	–0.840	*p* < 0.001
Visual processing	−901	*p* < 0.001	−832	*p* < 0.001	−601	0.001 *	–0.811	*p* < 0.001
Vestibular processing	−934	*p* < 0.001	−634	*p* < 0.001	−454	0.012 *	−737	*p* < 0.001
Touch processing	−913	*p* < 0.001	−699	*p* < 0.001	−504	0.004 *	–0.759	*p* < 0.001
Oral sensory	−818	*p* < 0.001	−732	*p* < 0.001	−507	0.003 *	–0.724	*p* < 0.001
Sensory inputs that influence emotional responses	−816	*p* < 0.001	−628	*p* < 0.001	−463	0.001 *	0.706	*p* < 0.001
Regulation of visual stimuli affecting emotional responses and activity level	−805	*p* < 0.001	−692	*p* < 0.001	−375	0.038 *	–0.704	*p* < 0.001
Emotional–social responses	−700	*p* < 0.001	−626	*p* < 0.001	−390	0.035 *	–0.718	*p* < 0.001
Behavioral consequences of sensory processing	−840	*p* < 0.001	−630	*p* < 0.001	−364	0.042 *	–0.718	*p* < 0.001
Response threshold results	−806	*p* < 0.001	−625	*p* < 0.001	−410	0.018*	–0.689	*p* < 0.001

SP: Sensory Profile; BPFAS: Behavioral Pediatric Feeding Assessment Scale; Spearman correlation analysis, * *p* < 0.05.

## Data Availability

The dataset analysed in this study can be requested from Mustafa Cemali (muscemali@hotmail.com) on reasonable request.

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
