# Peer review of "Relationship between Sensory Processing Skills and Feeding Behaviors in Children Aged 3–6 Years with Cerebral Palsy with Cerebral Visual Impairment"

_children, 2023, doi:10.3390/children10071188_

Round 1

Reviewer 1 Report

 As a reviewer of your article entitled "Relationship between Sensory Processing Skills and Feeding Behaviors in Children aged 3-6 years with Cerebral Palsy with Cerebral Visual Impairment  ", I am pleased to submit my comments and suggestions. Overall, I believe that your study addresses a relevant topic and provides valuable information on the relationship between sensory integration and feeding behavior in children with CP and visual impairment. However, I have also identified some issues that I feel need to be addressed to further strengthen your research and improve the quality of the article.

Below are my specific comments and suggestions:

- Please correct, in the key words, the word nutrition.

- Even if it has been included in the abstract, it would be advisable to include the meaning of the acronym CP the first time it appears in the manuscript (line 35)

- In the introduction, connectors should be used to link the different parts because some paragraphs are not connected  and it is difficult to follow the author's direction.

- write, please, in line 145, the 0 (0,87)

- In discussion section, other factors should be considered: Although sensory and feeding problems in children with CP and CVI are mentioned, other possible factors that could influence these results, such as severity of disability or the presence of comorbidities, are not explored in depth. It would be important to discuss how these factors might have influenced the results and to consider their possible interaction with the variables studied.

- The "limitations" in the discussion section is too short. It should be expanded more. The type of study, not being a clinical trial, has certain limitations to draw conclusive conclusions.

Author Response

Respond to Reviewer

Dear Reviewer

Thank you for your suggestions. Thanks to your suggestions, we were able to correct important deficiencies in the manuscript. We have tried to make all your revision suggestions meticulously. We tried to make the arrangements in line with your suggestions in order to ensure integrity, especially in the introduction and discussion section. We noticed that some parts were not understood because we had a problem with language translation. Thank you again for the details you have added to our work with your suggestions.

Kind regards

Revision

  1. Please correct, in the key words, the word nutrition.

Respond: “Nutrition” was removed from the keywords

  1. Even if it has been included in the abstract, it would be advisable to include the meaning of the acronym CP the first time it appears in the manuscript (line 35)

Respond: Necessary additions were made in line with your suggestion.

“Cerebral palsy (CP) is a non-progressive motor function, posture, and movement disability that occurs as a result of prenatal, natal, and postnatal harm to the developing brain (1).

  1. In the introduction, connectors should be used to link the different parts because some paragraphs are not connected and it is difficult to follow the author's direction.

Respond: Thank you for your suggestion. We noticed this inconsistency and deficiency in the introduction in line with your suggestions, and we rearranged the entire introduction in detail.

  1. write, please, in line 145, the 0 (0,87)

Respond: Önerileriniz doÄŸrultusunda gerekli düzeltme yapıldı.The total Cronbach α value of the scale is 0 (0.87) (31).”

  1. In discussion section, other factors should be considered: Although sensory and feeding problems in children with CP and CVI are mentioned, other possible factors that could influence these results, such as severity of disability or the presence of comorbidities, are not explored in depth. It would be important to discuss how these factors might have influenced the results and to consider their possible interaction with the variables studied.

Respond: The discussion section has been rearranged in line with your suggestions. We organized the entire section systematically to ensure overall integrity.

  1. The "limitations" in the discussion section is too short. It should be expanded more. The type of study, not being a clinical trial, has certain limitations to draw conclusive conclusions

Respond: Thank you for your valuable suggestion. Additions were made to the limitation in line with your suggestions.

“The study was limited in that it could not directly analyze sensory status or feeding beha-vior, as assessments were made using standard caregiver-filled questionnaires. In addi-tion, other limitations of the study are that other factors that may affect the feeding beha-vior such as the motor, swallowing and cognitive states of the children and the ability of the parents to feed the children were not evaluated.”

Reviewer 2 Report

1. The introduction fails to present solid evidence on the possible relationship between sensory processing and eating behaviors among CP children with CVI. Feeding behaviors among CP children may arise from the limited capacity of these children to tolerate various food textures and may not necessarily indicate underlying sensory processing issues.

2. End the introduction with a clear and explicit description of the study objectives.

3. Include the inclusion criteria for the mothers who accomplished the instruments.

4. Include in the materials and methods section a subsection on the procedures entailed in the study.

5. Indicate the mode of delivery (i.e., pen and paper, electronic) of the instrument.

6. For both instruments, include a short sentence (with appropriate reference/s) indicating that these have been previously used among children with CP.

7. Given the group differences between the three subgroups, why did the authors still conduct the correlation analysis as a whole instead of by each group?

8. Include the composite scores for the SP.

9. Is the CVI that may have influenced the SP scores, or could it have been the inherent feeding problems seen among children with CP that may explain this?

10. The conclusion needs to be framed to answer the objectives of the study directly.

The writing style needs to be improved. As a reader, I have difficulty understanding the writing, and it seems like it is directly translated from another language into English, without editing.

Author Response

Response to Reviewer

Dear Reviewer

Thank you for your suggestions. Thanks to your suggestions, we were able to correct important deficiencies in the manuscript. We have tried to make all your revision suggestions meticulously. While translating the text into English, we noticed that there were incorrect translations. We have done the language translation and editing much more carefully again. Thank you again for the details you have added to our work with your suggestions.

Kind regards

Revision

  1. The introduction fails to present solid evidence on the possible relationship between sensory processing and eating behaviors among CP children with CVI. Feeding behaviors among CP children may arise from the limited capacity of these children to tolerate various food textures and may not necessarily indicate underlying sensory processing issues.

Respond: Thank you for your suggestions. We noticed the shortcoming you mentioned in the introduction and fixed it. We rewrote the entire introduction.

  1. End the introduction with a clear and explicit description of the study objectives.

Respond: We rearranged the aim of the study by expressing it in a more regular way.

  1. Include the inclusion criteria for the mothers who accomplished the instruments.

Respond: Thank you for your suggestion, the necessary addition has been made to the participant section.

“Inclusion criteria for mothers; to be able to read and write, to be at a cognitive level capable of perceiving and answering the survey questions, and to agree to answer the survey ques-tions completely”

  1. Include in the materials and methods section a subsection on the procedures entailed in the study.

Respond: Thank you for your suggestion. We made the necessary adjustments and additions under the procedure title.

“2.2. Procedure”

Three independent groups consisting of mothers of children aged 3-6 years with CVI with CP, without CVI with CP and TD children were formed for the study. After the pur-pose of the study was explained to the mothers who voluntarily agreed to participate in the study, they were asked to sign the consent form. The interviews were made face to face and the evaluation forms were filled using paper and pencil. Demographic information form, Sensory Profile (SP) and Behavioral Pediatric Evaluation Scale (BPFAS) were used in the evaluations made in the presence of a physiotherapist and feeding and dietetics specia-list with 10 years of experience. Each participant's evaluation session lasted an average of 30 minutes. After calculating the evaluation scores for each group, the SP and BPFAS scores of the groups were compared. Correlation analysis was used separately for each group to examine the relationship between SP and BPFAS.”

  1. Indicate the mode of delivery (i.e., pen and paper, electronic) of the instrument.

Respond: We forgot to add this information, thank you for pointing this out. Information on how the evaluations are made in line with your suggestions has been added to the procedure section.

“The interviews were made face to face and the evaluation forms were filled using paper and pencil”

  1. For both instruments, include a short sentence (with appropriate reference/s) indicating that these have been previously used among children with CP.

Respond: In line with your suggestions, the information that it is used in children with CP for both CP and BPFAS has been added together with their references.

2.4.2. Sensory Profile (SP)

“The Sensory Profile is a frequently used questionnaire to evaluate sensory processing skills in children with CP (23).”

2.4.3. The Behavioral Pediatric Feeding Assessment Scale (BPFAS)

“The scale is a tool used to evaluate eating behavior in children with CP (10).”

  1. Given the group differences between the three subgroups, why did the authors still conduct the correlation analysis as a whole instead of by each group?

Respond: Thank you for your suggestion on this matter. Correlation analysis was performed separately within each group. We made the necessary additions to Table 5.

  1. Include the composite scores for the SP.

Respond: Thank you for your suggestion. Since the aim of our study was to compare sensory processing skills and eating behavior in children with CP with CVI and CP without CVI, we thought that it would be better not to include total SP scores because it did not include our aim.

  1. Is the CVI that may have influenced the SP scores, or could it have been the inherent feeding problems seen among children with CP that may explain this?

Respond: CVI is a diagnosis given in addition to the diagnosis of CP. The aim of our study was whether the sensory processing skills and eating status of children changed when CVI somtoms were added in addition to CP symptoms. As a result, we concluded that when CVI is added to CP symptoms, children's sensory processing skills and eating behaviors are more problematic. This is the answer to your question, that is, the sensory and feeding status of children with CP are worsened when the diagnosis of CVI is added. CVI affected feeding scores.

  1. The conclusion needs to be framed to answer the objectives of the study directly.

Respond: Thank you for your suggestion. In line with what you said, we rearranged the conclusion section in a simpler form that explains our purpose.

“It was observed that children with CP with CVI had more problematic sensory pro-cessing skills and feeding behavior attitudes than children with CP without CVI, and children with CP without CVI than children with TD. In addition, it was concluded that as the problems in sensory processing skills increased in each group, the feeding behavior problems also increased. As a result, it was observed that sensory processing skills and feeding behaviors of children with CP were affected, and with the addition of CVI diagnosis to the diagnosis of CP, the sensory and feeding status of children became worse.”

Round 2

Reviewer 2 Report

Overall, the authors were able to address my concerns. My only concern now is on the quality of writing.

As a final comment, I highly suggest professional editing services to improve writing quality and mechanics.